# Fingolimod (FTY720), a Sphinogosine-1-Phosphate Receptor Agonist, Mitigates Choroidal Endothelial Proangiogenic Properties and Choroidal Neovascularization

**DOI:** 10.3390/cells11060969

**Published:** 2022-03-11

**Authors:** Christine M. Sorenson, Mitra Farnoodian, Shoujian Wang, Yong-Seok Song, Soesiawati R. Darjatmoko, Arthur S. Polans, Nader Sheibani

**Affiliations:** 1Department of Pediatrics, School of Medicine and Public Health, University of Wisconsin, Madison, WI 53705, USA; cmsorenson@pediatrics.wisc.edu; 2McPherson Eye Research Institute, School of Medicine and Public Health, University of Wisconsin, Madison, WI 53705, USA; srdarjat@wisc.edu; 3Department of Ophthalmology and Visual Sciences, School of Medicine and Public Health, University of Wisconsin, Madison, WI 53705, USA; farnoodian@wisc.edu (M.F.); shoujianwang@wisc.edu (S.W.); song224@wisc.edu (Y.-S.S.); aspolans@wisc.edu (A.S.P.); 4Department of Cell and Regenerative Biology, School of Medicine and Public Health, University of Wisconsin, Madison, WI 53705, USA; 5Department of Biomedical Engineering, University of Wisconsin, Madison, WI 53705, USA

**Keywords:** age-related macular degeneration, inflammation, choroid, retinal pigment epithelium, retinal vasculature, microglia

## Abstract

Neovascular or wet age-related macular degeneration (nAMD) causes vision loss due to inflammatory and vascular endothelial growth factor (VEGF)-driven neovascularization processes in the choroid. Due to the excess in VEGF levels associated with nAMD, anti-VEGF therapies are utilized for treatment. Unfortunately, not all patients have a sufficient response to such therapies, leaving few if any other treatment options for these patients. Sphingosine-1-phosphate (S1P) is a bioactive lipid mediator found in endothelial cells that participates in modulating barrier function, angiogenesis, and inflammation. S1P, through its receptor (S1PR1) in endothelial cells, prevents illegitimate sprouting angiogenesis during vascular development. In the present paper, we show that, in choroidal endothelial cells, S1PR1 is the most abundantly expressed S1P receptor and agonism of S1PR1-prevented choroidal endothelial cell capillary morphogenesis in culture. Given that nAMD pathogenesis draws from enhanced inflammation and angiogenesis as well as a loss of barrier function, we assessed the impact of S1PR agonism on choroidal neovascularization in vivo. Using laser photocoagulation rupture of Bruch’s membrane to induce choroidal neovascularization, we show that S1PR non-selective (FTY720) and S1PR1 selective (CYM5442) agonists significantly inhibit choroidal neovascularization in this model. Thus, utilizing S1PR agonists to temper choroidal neovascularization presents an additional novel use for these agonists presently in clinical use for multiple sclerosis as well as other inflammatory diseases.

## 1. Introduction

Age-related macular degeneration (AMD) is the leading cause of vision loss among people aged 50 years and older in the United States and other industrialized nations. It causes damage to the center of the retina, which can interfere with simple everyday activities [1,2]. There are three stages of AMD: early, intermediate, and late. No effective treatments are available for early AMD, and effective treatments for the intermediate form are only marginally effective. There are two types of late AMD: the so-called dry and wet forms. In wet AMD, also called neovascular (nAMD) or exudative AMD, abnormal blood vessels grow underneath the retina. These vessels can leak fluid and blood, which may lead to swelling and damage to the central retina. The damage may be rapid and severe, and nAMD typically results in the most debilitating vision loss. There are no “cures” for dry AMD and the condition may progress even with current interventions [3]. The most common treatment for nAMD consists of intraocular injections of drugs to reduce the levels of vascular endothelial growth factor (VEGF). There are limitations and potential side effects associated with anti-VEGF therapy, including choroidal atrophy [4]. Thus, a need exists for new drugs to complement anti-VEGF agents or to act as stand-alone agents for the treatment of nAMD, a common and devastating disease that compromises vision in the growing population of older people.

Sphingosine-1-phosphate (S1P) is a bioactive lipid mediator produced by endothelial cells, platelets, and erythrocytes [5,6,7]. It has five high-affinity G-protein coupled receptors, S1PR1-5, and subtypes S1PR1-3 are important for vessel stability during development [8]. S1P plays a significant role in the modulation of angiogenesis, inflammation, and barrier integrity [9]. In fact, S1P is detected in the area of CNV in a laser-induced mouse CNV model [10,11], and intravitreal injections of a humanized monoclonal antibody that specifically binds S1P mitigates neovascularization [10]. To further support the importance of this pathway, an S1PR agonist (Fingolimod/FTY720) is clinically used for the treatment of relapsing multiple sclerosis [12]. FTY720 inhibits inflammation by blocking the expression of inflammatory cell adhesion molecules, including ICAM-1 and VCAM-1, and decreasing vascular permeability and the expression of proinflammatory cytokines by blocking NF-κB activity. Although the expression of S1PR in some retinal cells has been reported, their expression in the choroid requires further investigation. Thus, given the role S1P plays in endothelial cells, gaining a better appreciation of S1PR to modulate inflammation and pathologic angiogenesis in the choroid should provide novel treatment options for nAMD.

In the present study, we examine the expression of various S1PRs in mouse retinal and choroidal cells and showed that the major S1PR expressed in choroidal endothelial cells (ChECs) is S1PR1. We next addressed the relevance of the S1PR modulation for the proangiogenic properties of ChEC. ChEC incubated with the S1PR agonist FTY720 or its phosphorylated analog FTY720(S)-P was less migratory and had a diminished capacity to undergo capillary morphogenesis. We also show that incubation with CYM5442 (a S1PR1 selective agonist) significantly inhibits capillary morphogenesis further supporting a role for S1PR1, a receptor abundantly expressed in ChEC, in this process. To assess whether S1PR1 impacts angiogenesis in vivo, we utilize the mouse model of choroidal neovascularization (CNV), in which a laser is utilized to rupture Bruch’s membrane. Oral or intravitreal administration of FTY720 or CYM5442 to mice that underwent laser photocoagulation significantly inhibited CNV. Thus, the modulation of S1PR provides a novel strategy for the treatment of nAMD.

## 2. Material and Methods

### 2.1. Isolation and Characterization of Various Cells from Mouse Eyes

We previously described the isolation and culture of retinal vascular cells from Immorto mice, including endothelial cells (RECs), pericytes (RPCs), and astrocytes (RACs), using procedures established in our laboratory [13,14,15]. We also reported the isolation and characterization of retinal pigment epithelial (RPE) cells, choroidal endothelial cells (ChECs) and microglial cells (MGCs) [16,17,18]. More recently, we isolated choroidal pericytes (ChPCs), melanocytes (ChMCs), and mast cells from bone marrow (BMMC) using these mice (our unpublished data). The ChPCs were prepared and cultured in the same medium as previously described for RPCs [15]. The ChMCs were grown in Ham’s F10 medium (N6635; Sigma, St. Louis, MO) with 10% fetal bovine serum. The BMMCs were grown in pericyte medium containing murine recombinant IL-3 (213-13; 10 ng/mL) and SCF (250-03; 20 ng/mL) (Peprotech, Cranbury, NJ, USA). The identity and purity of all cells were confirmed by FACS analysis for the expression of specific markers and immunofluorescence staining with greater than 98% purity.

### 2.2. Expression of S1P Receptors

RNA was prepared from cells actively growing in 60 mm tissue culture plates (12556001; Fisher Scientific, Hanover Park, IL, USA). Total RNA was extracted using a combination of TRIzol reagent (15596026; Life Technologies, Grand Island, NY, USA) and an RNeasy mini kit (74104; Qiagen, Maryland, CA, USA) column for purification. The cDNA synthesis was performed from 1 μg of total RNA using the RNA to cDNA EcoDry Premix (Double Primed) kit (639549; Clontech, Mountain View, CA, USA). A 10-fold dilutions of cDNA was used as the template in qPCR assays, performed in triplicate on a Mastercycler Realplex (Eppendorf; Enfield, CT, USA) using the TB-Green Advantage qPCR Premix (639676; Clontech). The amplification conditions used were as follows: 95 °C for 2 min; 40 cycles of amplification (95 °C for 15 s, 60 °C for 40 s); and dissociation curve step (95 °C for 15 s, 60 °C for 15 s, 95 °C for 15 s). The linear regression line for nanograms of DNA was assessed from the relative fluorescent units (RFUs) at a threshold fluorescence value (Ct). Expression levels of target genes were quantified by comparing the RFU at the Ct to the standard curve and normalized by simultaneous amplification of 60S ribosomal protein L13α (Rpl13a), used as a housekeeping gene. The list of primers is provided in Table 1.

### 2.3. In Vitro Cell Viability

ChECs (3 × 10^3^ in 0.1 mL) were grown in 96-well plates (12556008; Fisher Scientific) overnight and then incubated with different concentrations of drug FTY720 (11975), FTY720 (S)-Phosphate (10006408), FTY720 (R)-Phosphate (10006407), and CYM5442 (16925; all from Cayman, Ann Arber, MI, USA), or DMSO (D8418; Sigma) in serum-containing medium for an additional 1–4 days. A cell titer 96 aqueous non-radioactive cell proliferation assay reagent (G5421; Promega, Madison, WI, USA) was added each day to a portion of the cells according to the manufacturer’s instructions. Absorbance was measured at a wavelength of 490 nm using a plate reader (Bio Tek, Santa Clara, CA, USA). The percent viability relative to control untreated cells is shown. Each experiment was repeated twice with at least two different isolations of cells.

### 2.4. In Vitro Cell Migration Assay

The bottom side of Transwell inserts (07200150; Fisher Scientific: 8 μm pore size, 6.5 mm membrane) were coated with fibronectin (CB40008, Fisher Scientific; 2 μg/mL in PBS) and left at 4 °C overnight. The Transwell inserts were washed with PBS, and blocked in PBS containing 1% BSA for 1 h at room temperature. Detached ChECs (trypsinized) were resuspended in serum-free DMEM medium. To 1 × 10^5^ cells/0.1 mL, varying concentrations of drug or DMSO were added and then placed on ice for 20 min after which the suspension was added to the top of the Transwell inserts. Serum-free medium (0.5 mL) was placed in each well of a 24-well plate (09761146, Fisher Scientific), and the inserts were then added and incubated for 4 h at 33 °C. Following the incubation, the cells that passed through the insert membrane were fixed with 2% paraformaldehyde (15710; Electron Microscopy Sciences, Hatfield, PA, USA) for 10 min at room temperature, stained with hematoxylin and eosin (H&E; HHS32 and HT110132, Sigma), and the inserts mounted on a slide. We counted 10 high-power fields (×200) to assess the number of cells that migrated. Each experiment employed triplicate samples.

### 2.5. In Vitro Capillary Morphogenesis Assays

Tissue culture plates (35 mm; 12556000, Fisher Scientific) were coated with 0.5 mL Matrigel (9 mg/mL; CB40234, BD Biosciences, Franklin Lakes, NJ, USA) and incubated at 37 °C for at least 30 min. ChECs (2 × 10^5^/2 mL) were placed in serum-free DMEM medium containing different concentrations of drug or DMSO and place on ice for 20 min. The cell suspension was then layered on top of Matrigel-coated plates and incubated at 37 °C. Fourteen hours later, photographs were taken with a Nikon microscope (ECLIPSE TS100; Nikon, Melville, NY, USA) in a digital format. For each condition, the branch points to 5 high-power fields (×100) to produce the mean number of branch points for each condition.

### 2.6. Animals

The animals utilized in these studies were used in accordance with our animal protocol that was reviewed and approved by the University of Wisconsin-Madison Animal Care and Use Committee. These studies are also in accordance with the ARVO Statement for the Use of Animals in Ophthalmic and Vision Research. We used 6-week-old wild-type C57BL/6j mice (10 per group) that were housed on a 12 h light–dark cycle. Food and water was available ad libitum.

### 2.7. Laser-Induced Choroidal Neovascularization (CNV)

The laser-induced CNV model was an important model that advanced our understanding of the pathogenesis of wet AMD. This model also afforded the development and testing of effective treatments. Developed in 1979 [19], it remains one of the commonly used models for wet AMD research. Laser photocoagulation-induced rupture of the Bruch’s membrane on day 0 was accomplished by anesthetizing mice with ketamine hydrochloride (100 mg/kg) and xylazine (10 mg/kg). The pupils were dilated using a drop of 1% tropicamide. Over the posterior pole of each eye, laser photocoagulation (75 μm spot size, 0.1 s duration, 120 mW) was performed in the 9, 12, and 3 o’clock positions. This was performed using a slit lamp delivery system of an OcuLight GL diode laser (Iridex, Mountain View, CA, USA) and to observe the retina, a handheld cover slip was used as a contact lens [20,21].

### 2.8. Drug Treatment

The mice were subjected to oral gavage 2 days before laser exposure and then daily gavage (5 days/week) for the following 2 weeks. The dose of FTY720 was 0.3 mg/kg diluted in water (0.1 mL). Control animals received water. As an alternative approach, mice were given an intravitreal injection of the desired drug at day 0 (1 μL of a 10 µM stock), which was experimentally found to be the most effective. Here, we utilized a pump microinjection apparatus and pulled glass micropipettes calibrated to deliver 1 µL of drug or vehicle (PLI100A; Harvard Apparatus, Holliston, MA, USA) for the intravitreal injections. Once the mice were anesthetized and the pupil dilated, the sharpened tip of the micropipette was passed through the sclera, just behind the limbus into the vitreous cavity under a dissecting microscope and the drug or vehicle delivered. Fourteen days following laser photocoagulation, the choroid/RPE was harvested and prepared for quantification of the area of neovascularization, as described below. For gene expression studies, the retinas and choroid/RPE tissues were dissected at different times post laser from untreated animals for RNA preparation and qPCR analysis, as detailed above.

### 2.9. Detection and Quantification of Choroidal Neovascularization

The eyes were fixed in 4% paraformaldehyde (4 °C for 2 h), washed 3 times in PBS, then sectioned at the equator, and the anterior half, the vitreous, and the retina were removed. The remaining choroid/RPE tissue was incubated in blocking buffer (20% fetal calf serum, 20% normal goat serum in PBS), then anti-ICAM-2 antibody (553326; BD Biosciences, Chicago, IL, USA) (1:500 in PBS containing 20% fetal calf serum, 20% normal goat serum) overnight at 4 °C. The choroid/RPE was washed and incubated with the appropriate secondary antibody. The choroid/RPE complex was dissected through five-to-six relaxing radial incisions and flat-mounted on a slide with VectaMount™ AQ (H550160, Vector Labs, Burlingame, CA, USA). All samples were visualized by fluorescence microscopy, and the images were captured in digital format using a Zeiss microscope. The area of CNV (in μm^2^) was analyzed with Image J software (National Institute of Mental Health, Bethesda, MD, USA; http://rsb.info.nih.gov/ij/, accessed on 20 September 2021).

### 2.10. Statistical Analysis

Here, we assessed statistical differences between 2 groups with the student’s unpaired *t*-test (two-tailed). We evaluated statistical differences between the groups with one-way ANOVA followed by Tukey’s Multiple Comparison Test using GraphPad Prism 8.0 (GraphPad Software, San Diego, CA, USA). We utilized Tukey’s Multiple Comparison Test to assess the significant differences between the means of every possible two groups in all experimental groups of three or more [22]. The mean ± SEM is shown. *p* < 0.05 was considered significant.

## 3. Results

### 3.1. S1PR1 Is the Highest-Expressing Subtype in Choroidal Endothelial Cells

S1P modulates angiogenesis and inflammation [9], both of which contribute to CNV. S1PR1-3 expression is widely reported in EC during development, while S1PR3-4 expression is more restricted [8]. To obtain a better understanding of the impact that these receptors have in the choroid, we assessed S1PR subtype expression in cultured murine ChECs as well as other ocular cell types and tissues.

We examined the expression of S1PR1-5 in retinal and choroidal cells by qPCR analysis of RNA isolated from these cells and retina and choroid/RPE tissues for comparison. The most robustly expressed S1PR in ChECs, such as REC, was S1PR1 (Figure 1A). S1PR5 was nearly undetectable in ChECs, while other family members, S1PR2, S1PR3, and S1PR4, had lower expressions. As a comparison, we further assessed S1PR expression in ChPC and RPE cells, as well as RPC, RAC, MGC, ChMC, and BMMC. ChPC, RPE cells, MGC, RAC, and ChMC, such as ChECs, expressed significantly higher levels of S1PR1 compared to other S1P receptors. RAC expressed mainly S1PR1 and S1PR2. This is consistent with astrocyte’s S1PR1 being a nonimmunological CNS target for FTY720 effectively targeting multiple sclerosis [23]. RPC had the most diverse S1PR expression, with S1PR4 most predominantly expressed followed by S1PR1and S1PR3. BMMC, such as RPC, predominantly expressed the S1PR4 receptor, but at nearly a 250-fold lower level than RPC, as well as low levels of other S1PRs. Similar to that observed in other ocular cell types, S1PR5 had the lowest expression of RPCs.

We next examined S1PR expression in the retina and choroid/RPE tissue before and at different times after laser photocoagulation (1, 3, and 30 days; Figure 1B). The S1PR expression patterns were very similar in retinas and choroid/RPE tissue from animals that did not undergo laser photocoagulation. These results are consistent with the expression patterns of S1PRs in retinal and choroidal cells indicating S1PR1 as the predominant receptor with a modest expression of other S1PRs, except for RPC and BMMC. Although laser photocoagulation had no significant effect on the expression of S1PR1 in the choroid/RPE tissue, it did significantly enhance S1PR1 expression in the retina. In the choroid/RPE, laser photocoagulation did not significantly affect S1PR3 expression. Although the expression of S1PR2 did not significantly change 1 day after laser photocoagulation, its level significantly decreased 3 and 30 days later. S1PR4 and S1PR5 expression increased 1 day following laser photocoagulation. S1PR5 expression was significantly higher in samples 30 days following laser photocoagulation. In the retina, S1PR5 expression was not affected by laser photocoagulation. S1PR2 and S1PR3 expression increased significantly 1 and 3 days following laser photocoagulation, while S1PR4 expression was significantly increased 3 and 30 days after laser photocoagulation. The expression of S1PR3, among S1P receptors, was most significantly upregulated in the retina at 1 and 3 days after laser photocoagulation, compared to controls, and returned to near-basal level by 30 days.

These results demonstrate that S1PR1 is abundantly expressed in ChECs and could play a predominant role in modulating their activity. Due to the important role S1P and its receptors play during vascular development and angiogenesis, we next set out to gain a better understanding of S1PR roles in the angiogenic properties of ChECs.

### 3.2. Decreased ChEC Viability with S1PR Agonists

First, we assessed the effect of FTY720 (a non-selective agonist for 4 of the 5 known S1P receptors, S1PR1,3-5), as well as CYM5442 (a S1PR1 selective agonist) on ChEC viability. The cells were grown with the solvent alone or agonist at varying concentrations. Figure 2A demonstrates that the incubation of ChECs with 10 µM FTY720 or CYM5442 significantly decreases the cell viability after 4 days. Thus, S1PR and S1PR1 agonists could negatively affect ChEC viability following long exposure.

The in vivo phosphorylation of FTY720 in rats and humans leads to the exclusive biologically active (S)-configured enantiomer [24]. Therefore, we assessed the impact of the phosphorylated form of the S1PR agonist FTY720 on ChEC viability. ChECs incubated with 10 µM phosphorylated FTY720(S)-P for 3 days demonstrated decreased viability, while the inactive enantiomer FTY720(R)-P had less impact on viability at the same concentration (Figure 2B,C). Thus, FTY720(S)-P similarly decreased ChEC viability.

### 3.3. Agonism of S1PR Inhibits ChEC Migration

The participation of EC during angiogenesis requires effective cell migration. Increased or decreased migration can greatly impact the ability of EC to undergo capillary morphogenesis. Here, we assessed the impact of S1PR agonism on ChEC migration by utilizing a Transwell migration assay. The incubation of ChECs with FTY720 (S1PR agonist) for 4 h significantly decreased ChEC migration (Figure 3). In contrast, incubation with CYM5442 (S1PR1 selective agonist) for 4 h had no significant impact on ChEC migration. Thus, another S1PR other than S1PR1 may be required for the modulation of ChEC migration. Next, we assessed whether the enantiomer of the phosphorylated S1PR agonist FTY720 modulated ChEC migration. Here, we show that FTY720(S)-P effectively inhibited ChEC migration (Figure 4). Thus, the agonism of S1PR decreased ChEC migration, a process critical for CNV.

### 3.4. FTY720 and CYM5442 Inhibit ChEC Capillary Morphogenesis

Capillary morphogenesis is an inherent characteristic of ECs necessary during normal vascular development and pathological neovascularization. To assess ChEC capillary morphogenesis, either with or without S1PR agonized by FTY720, ChECs were plated on Matrigel-coated plates for 14 h and imaged as previously reported [25,26]. FTY720 was very effective at preventing capillary morphogenesis. At 1 µM, incubation with FTY720 had a significant reduction in capillary morphogenesis. No visible capillary morphogenesis was noted with 5 and 10 µM FTY720 (Figure 5). We next assessed whether the phosphorylated FTY720 enantiomer could effectively inhibit capillary morphogenesis. Figure 6 demonstrates that 1 µM FTY720(S)-P effectively inhibits capillary morphogenesis. Thus, our data indicate an important role for S1PR activity in ChEC capillary morphogenesis.

Due to S1PR1 having the highest level of expression in ChECs, we next focused on the impact S1PR1 has during capillary morphogenesis. Agonizing S1PR1 in ChECs with increasing concentrations of CYM5442 (a selective S1PR1 agonist) resulted in a steady decline in the ability of these cells to undergo capillary morphogenesis (Figure 7). Thus, signaling through S1PR1 prevents capillary morphogenesis, as previously demonstrated [27].

### 3.5. Agonism of S1PR Inhibits Choroidal Neovascularization In Vivo

CNV results from the dysfunction of the choriocapillaris, which normally nourishes the outer retina [28,29,30]. Since S1PR agonists inhibited ChEC migration and capillary morphogenesis, we next assessed the impact of these agonists in the laser photocoagulation mouse model of CNV. C57BL/6j mice received vehicle or FTY720 by gavage prior to and following the laser photocoagulation-induced rupture of Bruch’s membrane or by intravitreal injection, as noted in Methods. To assess the CNV levels, anti-ICAM-2 wholemount staining of the RPE/choroid was performed 2 weeks following laser photocoagulation. Mice that received FTY720 by gavage (Figure 8) or intravitreal injection (Figure 9) showed a significant decrease in CNV.

Next, we assessed whether the selective agonism of S1PR1 was sufficient to decrease CNV in the mouse model following laser photocoagulation. The administration of the S1PR1 agonist CYM5442 intravitreally significantly reduced CNV (Figure 10). Thus, the modulation of S1PR1 activity plays an important role during pathologic neovascularization in the choroid.

## 4. Discussion

S1P acting as a phospholipid mediator modulates angiogenesis and inflammation through interactions with its receptors on various cell types. Although some studies investigated the role of S1P and its receptors in retinal vascular development and neovascularization, little is known about their role in choroidal vascular development and neovascularization. The pathogenesis of nAMD is driven by inflammation and aberrant angiogenesis leading to tortuous leaky choroidal vessels. The use of a humanized anti-S1P antibody was shown to block retinal and choroidal neovascularization in mice [10]. In addition, antagonism or lack of endothelium S1PR2, reported to block retinal neovascularization, and could impact the retinal outer-barrier integrity of RPE cells [9,31]. However, our unpublished results support a minimal role for S1PR2 in retinal vascular development and neovascularization during oxygen-induced ischemic retinopathy. S1PR2 expression is not significantly impacted by exposure to hyperoxia or hypoxia, while the expression of S1PR1 and S1PR3 is significantly upregulated by hyperoxia and downregulated by hypoxia. In fact, the retinal expression of S1PR1 and S1PR3, and not S1PR2, as predominant retinal S1P receptors is dramatically affected after laser treatment. During normal retinal vascular development, the expression of S1PR1 and S1PR3 is predominant, compared to other S1P receptors, and their expression is reciprocally regulated. With the completion of retinal vascularization, S1PR1 increases, while S1PR3 expression decreases. The details of molecular and cell-specific mechanisms of S1PR modulations and their impact on the retinal and choroidal vasculature deserve further investigation.

Endothelial cells are known as a major source of S1P [5,6,7]. S1P can act in an autocrine or paracrine manner impacting various cellular function. We first assessed the expression of S1P receptors in ChECs. We found that S1PR1 had substantially more expression in ChECs compared to other S1P receptors. We also examined the expression of S1P receptors in other ocular cell types from the retina and choroid. S1PR1 was predominant in REC, RAC, and RPE cells. S1PR1 expression was 5-to-10-fold higher in REC compared to ChEC, RPC, and RPE cells. S1PR1 expression was 25-fold lower in ChPCs, 40-fold lower in MGCs, 50-fold lower in RACs, 83-fold lower in ChMCs, and 2500-fold lower in BMMCs, compared to RECs. S1PR5 expression was very low in all the cells examined. S1PR2, S1PR3, and S1PR4 were moderately expressed in some cell types. RPCs showed the most varied expression of S1P receptors compared to other cell types, and S1PR4 was the predominant receptor in RPCs. The interaction of S1P with S1PR4 in neutrophils modulates 5-lipoxygenase activity with a significant impact on inflammatory processes [32]. The downstream activity of S1P/S1PR4 interactions in RPCs is unknown and may impact their function as important modulators of inflammatory processes in the central nervous system [33,34]. Thus, understanding the cell type-specific modulation of these receptors in ocular cells helps to advance our understanding of S1P receptor function during various ocular developmental and pathological processes.

S1PR1 expression in ECs is needed for perivascular-supporting cell coverage during vascular development [35]. S1PR1 controls the recruitment of perivascular-supporting cells to developing blood vessels with an integral role in vascular development [36]. Mice lacking S1PR1 die embryonically from severe bleeding, even though angiogenesis and vasculogenesis proceed normally, while mice deficient in S1PR2 or S1PR3 are viable and breed [36,37]. These findings are also supported by a study showing the importance of S1P production by RPCs and its interaction with S1PR1 on RECs [38]. These interactions are essential for the functional integrity of retinal vessels, whose dysregulation during diabetes leads to leaky and degenerative vasculature. Thus, coupled iterative chemistry and high-throughput screening could be used to identify S1P receptor and/or cell-specific compounds for the treatment of ocular diseases with a neovascular component.

The efficacy of FTY720 in a preclinical model of MS requires the modulation of S1PR1 on brain astrocytes [23]. In the limb, S1PR1 prevents VEGF-dependent and -independent sprouting angiogenesis [39]. The loss of S1PR1 in the embryonic hindbrain increases endothelial sprouting and ectopic branching [27]. These studies demonstrated that S1P signaling through S1PR1 squelches endothelial sprouting while stabilizing cell–cell adhesion. They also noted a decline of VEGF-induced signaling. Since hyperbranched aortas were observed in these studies, the investigators proposed that S1P through S1PR1 prevents the illegitimate sprouting and branching of endothelium in response to the VEGF [27]. Thus, it is tempting to speculate that the modulation of S1PR1 activity may aid in preventing pathologic VEGF-driven neovascularization and provide an important target for treatment of nAMD alone or in combination with anti-VEGF. Given the extensive toxicology and efficacy that studies have already performed with FTY720, and its recent analogs targeting a subset of S1PR subtypes [40] in humans with no sign of noted visual defects, make their repurposing for treatment of eye diseases immediately possible.

The modulation of S1PR signaling using S1P antibodies was shown to be effective in suppressing CNV [10,11] and vascular inflammation [41]. However, the modulation of S1PR in the choroid needs further investigation. The role S1PR1 plays in ChEC function has not been previously addressed. Here, we showed that the agonism of S1P receptors with FTY720 decreased the migration of ChECs and their capillary morphogenesis. Typically, we observed that altering the ECs’ migration rate detrimentally impacts their capillary morphogenesis in vitro [42,43]. Interestingly, here we observed diminished capillary morphogenesis of ChECs in the presence of CYM5442 (a S1PR1 selective agonist) with minimal impact on their migration. These results are consistent with the S1P signaling through S1PR1 suppressing sprouting angiogenesis through VE cadherin-dependent functions enhancing adherens junction formation and integrin activation stabilizing nascent blood vessels with efficient blood flow. Thus, these activities may explain the mitigation of neovascularization by CYM5442 without a direct effect on ChEC migration. CYM5442 is also shown to inhibit macrophage recruitment and reduce the severity of acute graft-versus-host disease. This is mediated through decreased production of proinflammatory mediators, including CCL2 and CCL7, by ECs and had a significant effect on inflammatory cell adhesion molecule expression on ECs [44], which impacts their proinflammatory activity and likely the mitigation of capillary morphogenesis without an impact on EC migration. This may also explain the reduced efficacy of CYM544, compared to FTY720, in the inhibition of CNV, as demonstrated in the present study.

The ability of FTY720 to block ChEC migration may occur through S1PR(s) other than S1PR1. S1PR1, S1PR3, and S1PR4 can promote migratory responses by Rac activation, while S1PR2 and S1PR5 could mitigate migration through Rho and Rac activation [45]. In addition, the inhibition of cell migration by S1PR1 activation could be the result of enhanced adherens junction assembly and integrin activation promoting vascular network stabilization [27], and, as a result, indirectly inhibit migration. Alternatively, the FTY720 effects could be mediated through the inhibition of Sphk1 and diminished levels of S1P [46]. S1P promotes cell survival and migration through S1P receptor-dependent and -independent pathways, and it is influenced by the cell type-specific expression of S1P receptors [45]. The potential involvement of other S1PR is supported by the FTY720 agonism of four of the five known S1P receptors (S1PR1 and S1PR3-5), which are also expressed in ChECs. In addition, complementary, non-overlapping roles are proposed for S1PR1-3 during retinal vascular development and neovascularization [8]. However, whether this is the case in the choroidal vascular development and neovascularization awaits further investigation. Thus, it is likely that the agonism of other S1PRs by FTY720, in addition to S1PR1, may mediate this anti-migratory activity in ChECs. Furthermore, the phosphorylated and active form of FTY720 can also act as a functional S1PR antagonist, where bound S1P receptors fail to recycle to the cell surface and instead are degraded [23]. This does not occur with the native S1P ligand and may explain the antagonist effect of S1P antibodies mitigating neovascularization [10,11]. The loss of S1PR1 results in increased sprouting angiogenesis and ectopic branching [27]. In contrast, the loss of S1PR2 may mitigate pathological retinal angiogenesis [31]. However, as shown here and indicated above, our results suggest that S1PR2 expression in the retina is very low compared to S1PR1 and S1PR3, and its expression is not dramatically affected by hyperoxia or hypoxia. Although the antagonism of S1PR1 could enhance angiogenesis, antagonism of S1PR2 prevents sprouting angiogenesis. Thus, our studies demonstrate a role for S1PR1, and perhaps other S1PRs in modulating ChEC sprouting, and their potential targeting in halting aberrant vascular sprouting.

Pathologic neovascularization is a major concern in many eye diseases, including nAMD. The only treatment modality presently available for nAMD patients is anti-VEGF, in which a subset of patients demonstrates an incomplete response. Given that nAMD pathologic angiogenesis is VEGF driven, the ability of S1PR1 to inhibit VEGF signaling and stabilize VE cadherin at endothelial junctions [27] makes it an attractive candidate for preserving vision by preventing CNV. Our data show the important role S1PR1 plays in suppressing CNV in vivo. Both FTY720 (a S1PR non-selective agonist), and CYM5442 (a S1PR1 selective agonist) to a lesser extent, suppressed CNV in the mouse model of the laser-induced Bruch’s membrane rupture. These results also support the importance of other S1P receptors (S1PR3-5) in the modulation of CNV. However, the exact contribution of these S1P receptors awaits their future targeted studies. Thus, our findings not only point to potential new uses for a drug already in use, but also to the importance of identifying the mechanisms necessary to protect the vasculature from unrestrained angiogenic sprouting, either in the mature vasculature or during development.

## Figures and Tables

**Figure 1 cells-11-00969-f001:**
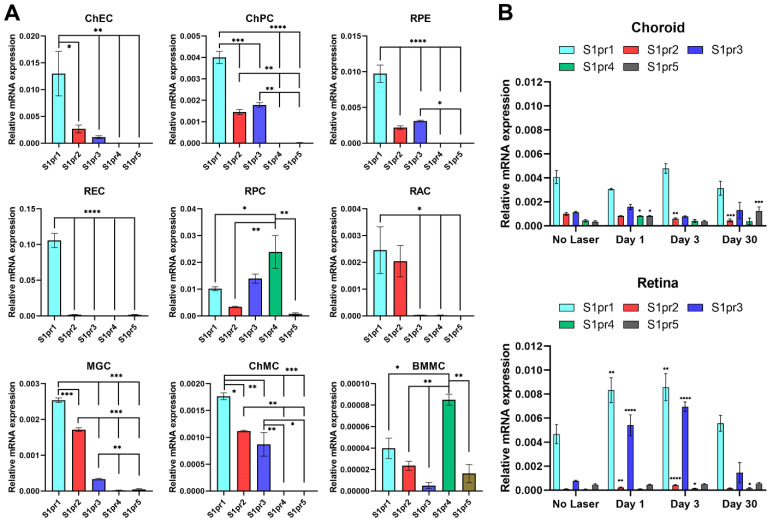
Expression of S1PR in various ocular cells. (**A**) Total RNA was extracted from murine ChEC, ChPC, RPE, REC, RPC, RAC, MGC, ChMC, and BMMC or (**B**) retina and choroid/RPE tissue, as outlined in Methods. A total of 1 µg of total RNA was used for cDNA synthesis. cDNA was used as the template for qPCR performed in triplicates of three biological replicates using specific gene primers (Table 1). The RpL13A was used as a control. The relative amounts of transcripts were determined, as detailed in Methods. * *p* < 0.05; ** *p* < 0.01, *** *p* < 0.001, **** *p* < 0.0001. (*n* = 3).

**Figure 2 cells-11-00969-f002:**
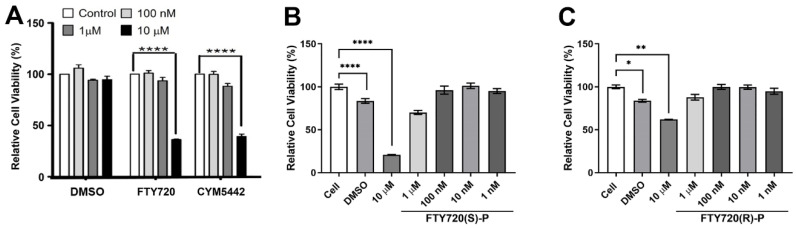
The S1PR non-selective agonist FTY720 and the S1PR1 selective agonist CYM5442 impact ChEC viability. (**A**) ChECs were grown in 96-well plates and incubated with medium containing varying concentrations of FTY720 or CYM5442 (in DMSO) or DMSO for 4 days. (**B**,**C**) ChECs were also incubated with different concentrations of active (FTY720(S)-P) or inactive (FTY720(R)-P) FTY720 enantiomers for 2 days. The percent cell viability relative to control samples was determined as described in Methods. * *p* < 0.05; ** *p* < 0.01, **** *p* < 0.0001. (*n* = 3).

**Figure 3 cells-11-00969-f003:**
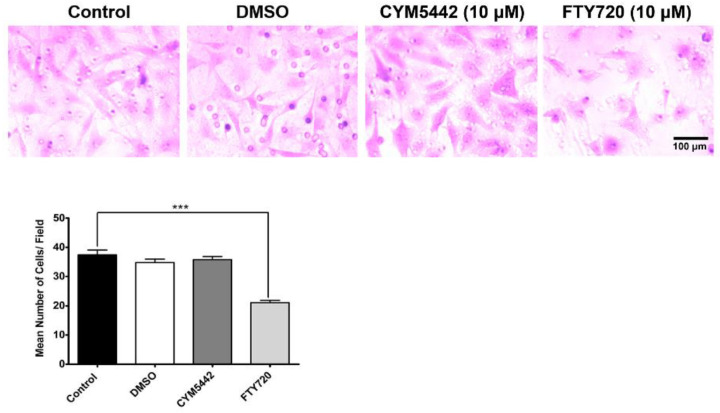
The S1PR agonist FTY720 inhibits ChEC migration. A suspension of ChECs was prepared in serum-free medium and added to the top of Transwell inserts and incubated for 4 h with medium containing agonist (in DMSO) or DMSO. Cells were fixed, stained with H&E, and the number of cells that migrated through the 8 μm pore size membrane was counted, as described in Methods. *** *p* < 0.001. (*n* = 3).

**Figure 4 cells-11-00969-f004:**
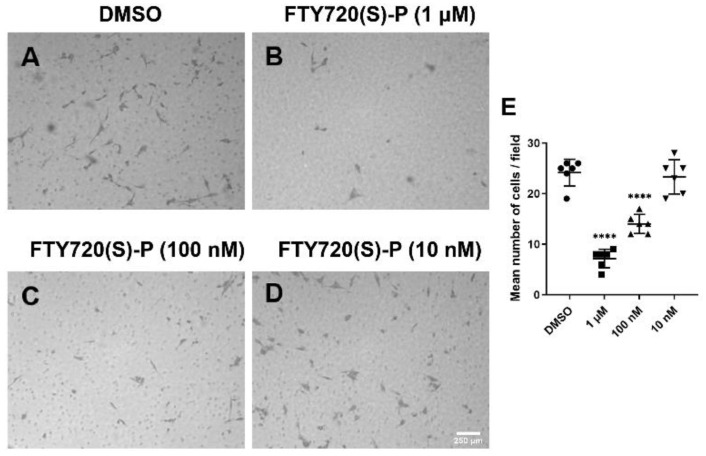
Enantiomers of the phosphorylated S1PR agonist FTY720 inhibit ChEC migration. ChECs in suspension were added to the top of Transwell inserts and incubated for 4 h with medium containing varying concentrations of enantiomer (in DMSO) or DMSO. Cells were fixed, stained with H&E, and the number of cells that migrated through the 8 μm pore size membrane was counted as described in Methods. **** *p* < 0.0001. (*n* = 3).

**Figure 5 cells-11-00969-f005:**
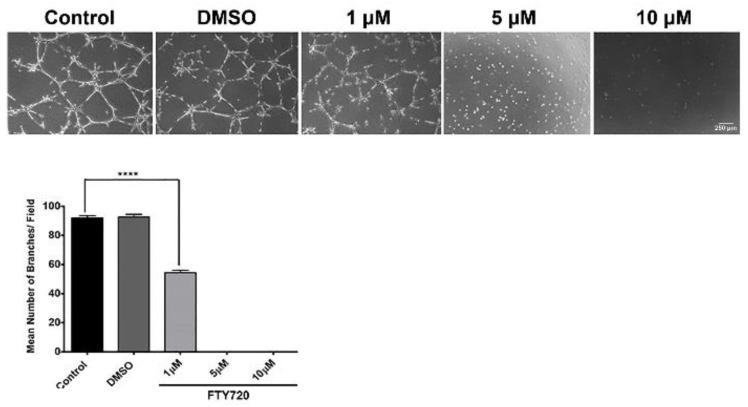
The S1PR agonist FTY720 inhibits ChEC capillary morphogenesis. Detached ChECs were incubated with varying concentrations of FTY720 (in DMSO) or DMSO on ice for 20 min. Incubated cells were then applied to Matrigel-coated plates, incubated, and imaged 14 h later as described in Methods. The mean number of branch points was determined as detailed in Methods. **** *p* < 0.0001. (*n* = 3).

**Figure 6 cells-11-00969-f006:**
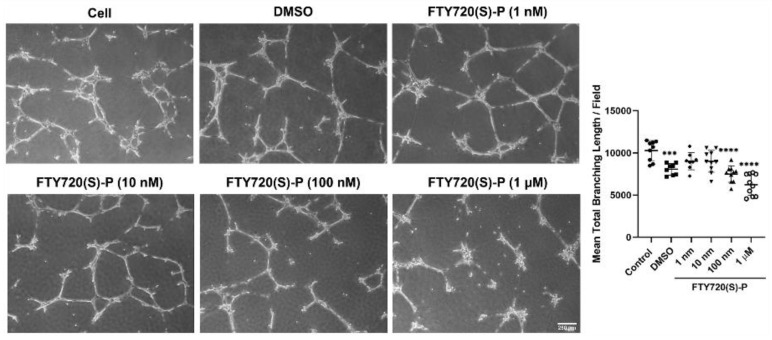
An enantiomer of the phosphorylated S1PR agonist FTY720 inhibits ChEC capillary morphogenesis. Detached ChECs were incubated with varying concentrations of phosphorylated S1PR agonist (in DMSO) or DMSO on ice for 20 min. Incubated cells were then applied to Matrigel-coated plates, incubated, and photographed after 14 h as described in Methods. The mean number of branch points was determined. *** *p* < 0.001, **** *p* < 0.0001. (*n* = 3).

**Figure 7 cells-11-00969-f007:**
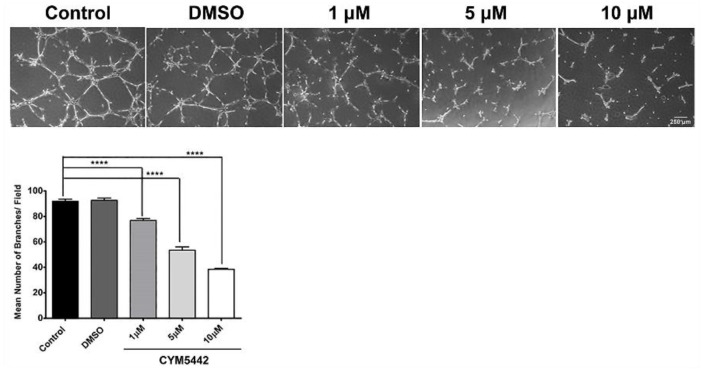
The S1PR1 agonist CYM5442 inhibits ChEC capillary morphogenesis. ChECs were detached and incubated with varying concentrations of CYM5442 (in DMSO) or DMSO on ice for 20 min. Incubated cells were then applied to Matrigel-coated plates, incubated, and photographed after 14 h as described in Methods. The mean number of branch points was determined. **** *p* < 0.0001. (*n* = 3).

**Figure 8 cells-11-00969-f008:**
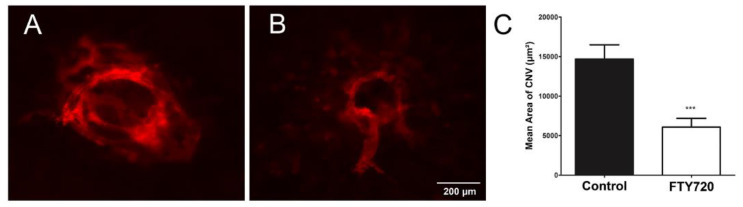
Oral delivery of the S1PR agonist FTY720 inhibits CNV in a mouse model of nAMD. Vehicle control (**A**) or FTY720 (**B**) was administered orally (0.3 mg/kg body weight) to C57BL/6j male and female mice (2 months old), 2 days before laser photocoagulation and for 2 weeks (5 times per week), as described in Methods. The concentration was chosen to mimic the safe and effective dose frequently used to treat other disease entities in mouse models. Representative images of new choroidal blood vessels stained with anti-ICAM-2 antibodies are shown in Panels A and B, and the quantification of the findings is shown in (**C**). *** *p* < 0.001. (*n* = 10 mice per group).

**Figure 9 cells-11-00969-f009:**
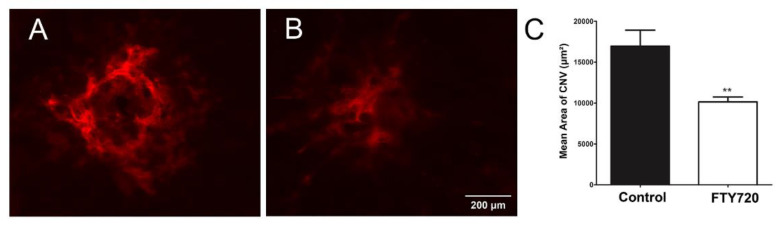
A single intravitreal injection of the S1PR agonist FTY720 inhibits CNV in a mouse model of nAMD. C57BL/6j male and female mice (2 months old) were injected intravitreally with (**A**) vehicle control or (**B**) FTY720, 1 μL of a 10 µM stock, following a laser rupture of Bruch’s membrane as described in Methods. After 14 days, the eyes were processed for staining with anti-ICAM-2 antibodies and quantification of new choroidal blood vessels (**C**). ** *p* < 0.01. (*n* = 10 mice per group).

**Figure 10 cells-11-00969-f010:**
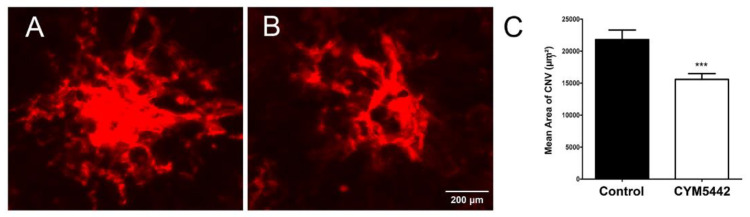
A single intravitreal injection of the S1PR1 agonist CYM5442 inhibits CNV in a mouse model of nAMD. C57BL/6j male and female mice, 2 months old, received a single intravitreal injection of (**A**) vehicle control or (**B**) CYM5442, 1 μL of a 10 µM stock, following the laser rupture of Bruch’s membrane as described in Methods. After 14 days, the eyes were processed for staining with anti-ICAM-2 antibodies and quantification of new choroidal blood vessels (**C**). *** *p* < 0.001. (*n* = 10 mice per group).

**Table 1 cells-11-00969-t001:** List of mouse primers used in qPCR analysis.

Gene	Forward (5′ to 3′)	Reverse (5′ to 3′)
**S1pr1**	CGGTGTAGACCCAGAGTCCT	AGCTTTTCCTTGGCTGGAG
**S1pr2**	CCCAACTCCGGGACATAGA	ACAGCCAGTGGTTGGTTTTG
**S1pr3**	TCTCCCAATTGTTCCCTGAA	ACTTCAACAGTCCACGAGAGG
**S1pr4**	CGTGATGAATGTTTGGCAGA	CCCTTCAAGGCCCAGACT
**S1pr5**	CAAGACTTCCCCACAACCTG	TATGGCTGCAGCAGAAATTG
**Rpl13a**	TCTCAAGGTTGTTCGGCTGAA	CCAGACGCCCCAGGTA

## Data Availability

All the data in support of the findings presented are included here and can be shared by the corresponding author with reasonable requests.

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
