# Peer review of "Fingolimod (FTY720), a Sphinogosine-1-Phosphate Receptor Agonist, Mitigates Choroidal Endothelial Proangiogenic Properties and Choroidal Neovascularization"

_cells, 2022, doi:10.3390/cells11060969_

Round 1
Reviewer 1 Report
S1P1 receptor agonist mitigation of choroidal endothelial proangiogenic properties and choroidal neovascularisation is a concept that this reviewer has concluded from the literature and therefore a manuscript which has experimentally demonstrated this is very favourably received by this reviewer. The manuscript is a very important addition and is novel with important implications for future therapeutic application.
There are a number of minor revisions that should be addressed by the authors in the manuscript. These are:
- The effects of FTY720 and CYM on ChEC viability is a little concerning in terms of potential therapeutic treatment and some discussion is required here-especially as FTY720 (parent molecule of FTY720 phosphate) inhibits SphK1 (cell survival protein) at high concentrations and this could lead to reduced cell viability. The authors should also clearly highlight that effects on cell migration and capillary morphology are measured at a shorter time compared with the impact on cell viability (4 days).
-
Some information is needed to dissect the difference in effects of CYM and FTY720 phosphate on ChEC migration. Is this due to a differential balance effect of agonism versus functional antagonism for the two ligands? This is an alternative explanation to invoking other S1P receptor sub types being involved in the CYM effect. Afterall, S1P1 receptor agonism should promote migration, and functional antagonism should inhibit.
3. The impact of the lack of effect on migration and the inhibition of capillary morphogenesis by CYM suggests that migration is disconnected for CYM. Is the agonism of S1P1 by CYM and FTY720 phosphate inducing differentiation of vessels to consequentially inhibit spouting/branching?-therefore independent of effects on migration in this case.
4. The authors should also consider S1P receptor modulation combined with anti-VEGF treatment in the discussion. Any data on this would also enhance the impact of the manuscript if available.
5. How viable is the experimental model in terms of human disease-some commentary is warranted here to inform the reader.
Author Response
I would like to thank the reviewer for careful evaluation of our manuscript. The comments and responses are listed below:
S1P1 receptor agonist mitigation of choroidal endothelial proangiogenic properties and choroidal neovascularization is a concept that this reviewer has concluded from the literature and therefore a manuscript which has experimentally demonstrated this is very favorably received by this reviewer. The manuscript is a very important addition and is novel with important implications for future therapeutic application.
Thank you for the positive response, and the appreciation for the significance of the work presented here.
There are a number of minor revisions that should be addressed by the authors in the manuscript. These are:
- The effects of FTY720 and CYM on ChEC viability is a little concerning in terms of potential therapeutic treatment and some discussion is required here-especially as FTY720 (parent molecule of FTY720 phosphate) inhibits SphK1 (cell survival protein) at high concentrations and this could lead to reduced cell viability. The authors should also clearly highlight that effects on cell migration and capillary morphology are measured at a shorter time compared with the impact on cell viability (4 days).
We agree and highlight the time of exposure differences in these different experiments. The does responses were used to determine the levels of toxicity. For functional studies lower doses and/or incubation times were used to separate the impacts on viability from various activities. We now highlight the different lengths of treatments in the experiments conducted and point out other activities of FTY720.
- Some information is needed to dissect the difference in effects of CYM and FTY720 phosphate on ChEC migration. Is this due to a differential balance effect of agonism versus functional antagonism for the two ligands? This is an alternative explanation to invoking other S1P receptor sub types being involved in the CYM effect. Afterall, S1P1 receptor agonism should promote migration, and functional antagonism should inhibit.
Published studies have demonstrated that FTY720 interacts with multiple S1P receptors while CMY is specific for S1P1. We agree the alternative explanation is also feasible and now included. However, CMY agonism of S1P1, as demonstrated by S1P, should suppress vascular sprouting, a VE-cadherin dependent mechanism, inducing AJ assembly and integrin activation in order to stabilize nascent blood vessels with efficient flow. Thus, S1PR1 agonism should mitigate migration and functional antagonism should enhance migration as occur in S1PR1 deficient mice [1-3]. In contrast, S1PR2 expression is essential for angiogenesis [4]. In addition, S1PR1-3 function coordinately during angiogenesis in a non-overlapping manner [5]. This also support the significance of engaging more than one S1PR by the agonist and its impact on angiogenesis. These comments are now incorporated into the discussion.
- The impact of the lack of effect on migration and the inhibition of capillary morphogenesis by CYM suggests that migration is disconnected for CYM. Is the agonism of S1P1 by CYM and FTY720 phosphate inducing differentiation of vessels to consequentially inhibit spouting/branching? -therefore independent of effects on migration in this case.
This notion is supported by the proposed function of S1PR1 activity in mediating junctional stability of endothelium and reducing vascular permeability and could potentially be indirectly impacting migration. This is now included in the discussion.
- The authors should also consider S1P receptor modulation combined with anti-VEGF treatment in the discussion. Any data on this would also enhance the impact of the manuscript if available.
This is a good point and unfortunately, we did not look at the combined effects of anti-VEGF and S1P agonism in our in vivo model. Since S1P signaling via S1PR1 suppresses VEGF dependent vascular sprouting, in a mechanism dependent on VE-cadherin function, one anticipates potential additive/synergistic effects, which deserves future investigation. In fact, a recent study showed that coordinated VEGF and S1P signaling establishes a gradient of JunB/AP1, which establishes normal vascularization. VEGF signaling drives the expression of JunB which is attenuated by S1P-S1PR-depednet AJ assembly and barrier function, promoting normal blood flow [6].
- How viable is the experimental model in terms of human disease-some commentary is warranted here to inform the reader.
This model was instrumental in the development of existing treatments for nAMD, i.e preclinical testing of anti-VEGF, as well as others. Thus, laser induced CNV model is a valuable preclinical model for testing the targeting of desired pathways, especially the power of genetic manipulation, to mitigate neovascularization in the choroid. This is now added to the discussion.
References:
- Gaengel, K.; Niaudet, C.; Hagikura, K.; Laviña, B.; Muhl, L.; Hofmann, J.J.; Ebarasi, L.; Nyström, S.; Rymo, S.; Chen, L.L., et al. The sphingosine-1-phosphate receptor s1pr1 restricts sprouting angiogenesis by regulating the interplay between ve-cadherin and vegfr2. Dev Cell 2012, 23, 587-599.
- Ben Shoham, A.; Malkinson, G.; Krief, S.; Shwartz, Y.; Ely, Y.; Ferrara, N.; Yaniv, K.; Zelzer, E. S1p1 inhibits sprouting angiogenesis during vascular development. Development 2012, 139, 3859-3869.
- Akhter, M.Z.; Chandra Joshi, J.; Balaji Ragunathrao, V.A.; Maienschein-Cline, M.; Proia, R.L.; Malik, A.B.; Mehta, D. Programming to s1pr1(+) endothelial cells promotes restoration of vascular integrity. Circ Res 2021, 129, 221-236.
- Skoura, A.; Sanchez, T.; Claffey, K.; Mandala, S.M.; Proia, R.L.; Hla, T. Essential role of sphingosine 1-phosphate receptor 2 in pathological angiogenesis of the mouse retina. J. Clin. Invest. 2007, 117, 2506-2516.
- Kono, M.; Mi, Y.; Liu, Y.; Sasaki, T.; Allende, M.L.; Wu, Y.P.; Yamashita, T.; Proia, R.L. The sphingosine-1-phosphate receptors s1p1, s1p2, and s1p3 function coordinately during embryonic angiogenesis. J. Biol. Chem. 2004, 279, 29367-29373.
- Yanagida, K.; Engelbrecht, E.; Niaudet, C.; Jung, B.; Gaengel, K.; Holton, K.; Swendeman, S.; Liu, C.H.; Levesque, M.V.; Kuo, A., et al. Sphingosine 1-phosphate receptor signaling establishes ap-1 gradients to allow for retinal endothelial cell specialization. Dev Cell 2020, 52, 779-793.e777.
Reviewer 2 Report
REVIEWER’S COMMENTS
The manuscript “Fingolimod (FTY720), a Sphinogosine-1-Phosphate Receptor Agonist, Mitigates Choroidal Endothelial Proangiogenic Properties and Choroidal Neovascularization” by Sorenson et al demonstrates that S1PR non-selective (FTY720) and S1PR1 selective (CYM5442) agonists significantly inhibited choroidal neovascularization.
- Please improve the resolution of images. The resolution should be at least 300 dpi.
- Please include the sample size i.e., “n” in the figure legends/ results.
- Please add the catalog numbers of all kits, antibodies, and reagents used in the “materials and methods” section.
- Please be consistent with the style of references.
Author Response
We thank the reviewer for careful evaluation of our manuscript. The comments and responses are shown below.
The manuscript “Fingolimod (FTY720), a Sphinogosine-1-Phosphate Receptor Agonist, Mitigates Choroidal Endothelial Proangiogenic Properties and Choroidal Neovascularization” by Sorenson et al demonstrates that S1PR non-selective (FTY720) and S1PR1 selective (CYM5442) agonists significantly inhibited choroidal neovascularization.
- Please improve the resolution of images. The resolution should be at least 300 dpi. They are made at least at 300 dpi.
- Please include the sample size i.e., “n” in the figure legends/ results. Done.
- Please add the catalog numbers of all kits, antibodies, and reagents used in the “materials and methods” section. Done.
- Please be consistent with the style of references. We have used the journal’s endnote style.